# Nodal β spectrins are required to maintain Na+ channel clustering and axon integrity

Cheng-Hsin Liu[1,2], Sharon R Stevens[1], Lindsay H Teliska[1], Michael Stankewich[3], Peter J Mohler[4], Thomas J Hund[5], Matthew N Rasband[1,2]*

[1]Department of Neuroscience, Baylor College of Medicine, Houston, United States; [2]Program in Developmental Biology, Baylor College of Medicine, Houston, United States; [3]Department of Pathology, Yale University, New Haven, United States; [4]Department of Physiology and Cell Biology, The Ohio State University, Columbus, United States; [5]Biomedical Engineering, The Ohio State University, Columbus, United States

**Abstract** Clustered ion channels at nodes of Ranvier are critical for fast action potential propagation in myelinated axons. Axon-glia interactions converge on ankyrin and spectrin cytoskeletal proteins to cluster nodal Na+ channels during development. However, how nodal ion channel clusters are maintained is poorly understood. Here, we generated mice lacking nodal spectrins in peripheral sensory neurons to uncouple their nodal functions from their axon initial segment functions. We demonstrate a hierarchy of nodal spectrins, where β4 spectrin is the primary spectrin and β1 spectrin can substitute; each is sufficient for proper node organization. Remarkably, mice lacking nodal β spectrins have normal nodal Na+ channel clustering during development, but progressively lose Na+ channels with increasing age. Loss of nodal spectrins is accompanied by an axon injury response and axon deformation. Thus, nodal spectrins are required to maintain nodal Na+ channel clusters and the structural integrity of axons.

*For correspondence:
rasband@bcm.edu

**Competing interests:** The authors declare that no competing interests exist.

## Introduction

Rapid and efficient saltatory action potential conduction depends on myelin and clustered ion channels at nodes of Ranvier. Disruption of nodes occurs in many neurological diseases and injuries (*Craner et al., 2004*; *Nashmi and Fehlings, 2001*; *Susuki, 2013*), and discovery of the basic mechanisms that cluster and maintain nodal ion channels may lead to therapeutic approaches to repair, preserve, and rebuild nodes. Previous studies revealed that two distinct neuron-glia interactions converge on the axonal cytoskeleton to cluster nodal ion channels during development (*Amor et al., 2017*; *Feinberg et al., 2010*; *Susuki et al., 2013*; *Zonta et al., 2008*). However, the mechanisms that maintain nodes throughout life are much less understood. Previous studies showed that loss of nodal extracellular matrix (ECM) molecules results in the gradual reduction of nodal voltage-gated Na+ (Nav) channel density (*Amor et al., 2014*). In addition, the AnkG-binding and node-enriched cytoskeletal protein β4 spectrin has been proposed to stabilize nodes (*Berghs et al., 2000*; *Komada and Soriano, 2002*; *Lacas-Gervais et al., 2004*; *Yang et al., 2004*). Thus, like node assembly, node maintenance may depend on both extrinsic and intrinsic interactions.

We recently reported human patients with pathogenic variants of *SPTBN4*, the gene encoding β4 spectrin (*Wang et al., 2018*). These patients have severe intellectual disability, hypotonia and motor axonal neuropathy. Similarly, mice lacking β4 spectrin have ataxia, reduced lifespan, and reduced nodal Nav channel density (*Berghs et al., 2000*; *Komada and Soriano, 2002*; *Lacas-Gervais et al., 2004*; *Yang et al., 2004*). Together, these studies showed that β4 spectrin is essential for nervous system function and may contribute to maintenance of nodal Nav channel clusters. However, these patients and mice have normal peripheral sensory axon function (*Wang et al., 2018*; *Yang et al.,*

2004), and the percentage of nodes with Nav channels is unchanged (*Ho et al., 2014*; *Susuki et al., 2013*). We found that βI spectrin, the major β spectrin in erythrocytes, is located at nodes in β4 spectrin mutant mice (*Ho et al., 2014*). Thus, β1 spectrin may partially compensate for loss of β4 spectrin to maintain nodal Nav channels and sensory axon function. How can the preserved sensory axon function be reconciled with the widespread neurological symptoms found in humans and mice with pathogenic β4 spectrin variants? These differences may reflect disruption of axon initial segments (AIS), rather than nodes of Ranvier. AIS are highly enriched with β4 spectrin and function to both initiate action potentials and regulate the proper trafficking and sorting of somatodendritic and axonal cargoes (*Leterrier, 2018*). Thus, to determine the function of nodal spectrins, it is necessary to uncouple their role at the AIS from their role at nodes. Furthermore, since β1 and β4 spectrin can compensate for each other at nodes, it is necessary to remove both spectrins simultaneously to determine the function of the nodal spectrin cytoskeleton. Pseudounipolar sensory neurons in the dorsal root ganglia in vivo may be an excellent experimental system to test the function of nodal β spectrins since many are myelinated with nodes of Ranvier, but most lack an AIS (*Gumy et al., 2017*).

We previously generated mice lacking α2 spectrin, the only neuronal α spectrin and obligate binding partner for β1 and β4 spectrin, in sensory neurons. These mice had profound defects in node of Ranvier assembly and showed axon degeneration (*Huang et al., 2017b*). However, these pathologies cannot be ascribed solely to loss of nodal spectrins since α2 spectrin is found throughout the neuron and not only at nodes (*Huang et al., 2017a*).

To determine the function of the nodal spectrin cytoskeleton, we generated β1 and β4 spectrin single conditional knockout mice, and β1/β4 spectrin double conditional knockout mice. We uncoupled their AIS and node functions by using *Avil-cre* mice to specifically remove these spectrins from peripheral sensory neurons. We found that although β4 spectrin is the primary nodal spectrin, in its absence β1 spectrin can fully substitute. Remarkably, mice lacking both β1 and β4 spectrin have normal Nav channel clustering during node assembly. However, loss of nodal spectrins causes the progressive loss of nodal Nav channels and neuronal injury with increasing age. These results finally demonstrate that the nodal spectrin cytoskeleton is required to maintain, but not assemble, nodal Nav channel clusters, and that disruption of the nodal cytoskeleton alone is sufficient to induce an axon injury response.

## Results

### β4 spectrin is dispensable for nodal Nav channel clustering

To disrupt the function of β4 spectrin in axons we used a conditional null allele with exons 29 and 30 of the mouse *Sptbn4* gene flanked by *loxP* sites (*Sptbn4$^{F/F}$* mice; *Unudurthi et al., 2018*). In the presence of Cre recombinase exons 29 and 30 are excised; the excision of exons 29 and 30 is predicted to disrupt both β4Σ1 and β4Σ6 spectrin splice variants, the two major forms of β4 spectrin found at AIS and nodes of Ranvier (*Komada and Soriano, 2002*; *Lacas-Gervais et al., 2004*; *Uemoto et al., 2007*; *Yoshimura et al., 2016*). We confirmed loss of β4Σ1 and β4Σ6 spectrin splice variants by crossing *Sptbn4$^{F/F}$* mice with *Nestin-Cre* mice (*Nes-Cre*). Immunoblotting of brain homogenate with an antibody generated against the C-terminal SD domain found in β4Σ1 and β4Σ6 spectrin splice variants showed that both variants are lost in *Nes-Cre;Sptbn4$^{F/F}$* mice (*Figure 1A*). To further confirm the loss of AIS β4 spectrin and to verify that no truncated N-terminal fragments of β4Σ1 are at the AIS, we used both C- and N-terminal-directed β4 spectrin antibodies to immunolabel cortical neurons in *Sptbn4$^{F/F}$* and *Nes-Cre;Sptbn4$^{F/F}$* mice; we found no AIS immunoreactivity in *Nes-Cre;Sptbn4$^{F/F}$* mice (*Figure 1B*; note, the nuclear signal detected with the N-terminal-directed antibody is not specific to β4 spectrin). These results show that Cre-dependent recombination in *Sptbn4$^{F/F}$* mice effectively eliminates both β4Σ1 and β4Σ6 spectrin splice variants.

To determine the function of β4 spectrin at nodes of Ranvier, and to circumvent the confound of loss of β4 spectrin from AIS, we crossed *Sptbn4$^{F/F}$* mice to *Advillin-Cre* (*Avil-Cre*) mice to specifically eliminate β4 spectrin from peripheral sensory neurons. This strategy also enables a direct comparison between sensory dorsal roots (no β4 spectrin) and motor ventral roots (control) in the same mouse. To determine if proprioception (mainly mediated by myelinated axons with nodes) or nociception (mainly mediated by unmyelinated axons lacking nodes) are altered in *Avil-Cre; Sptbn4$^{F/F}$*

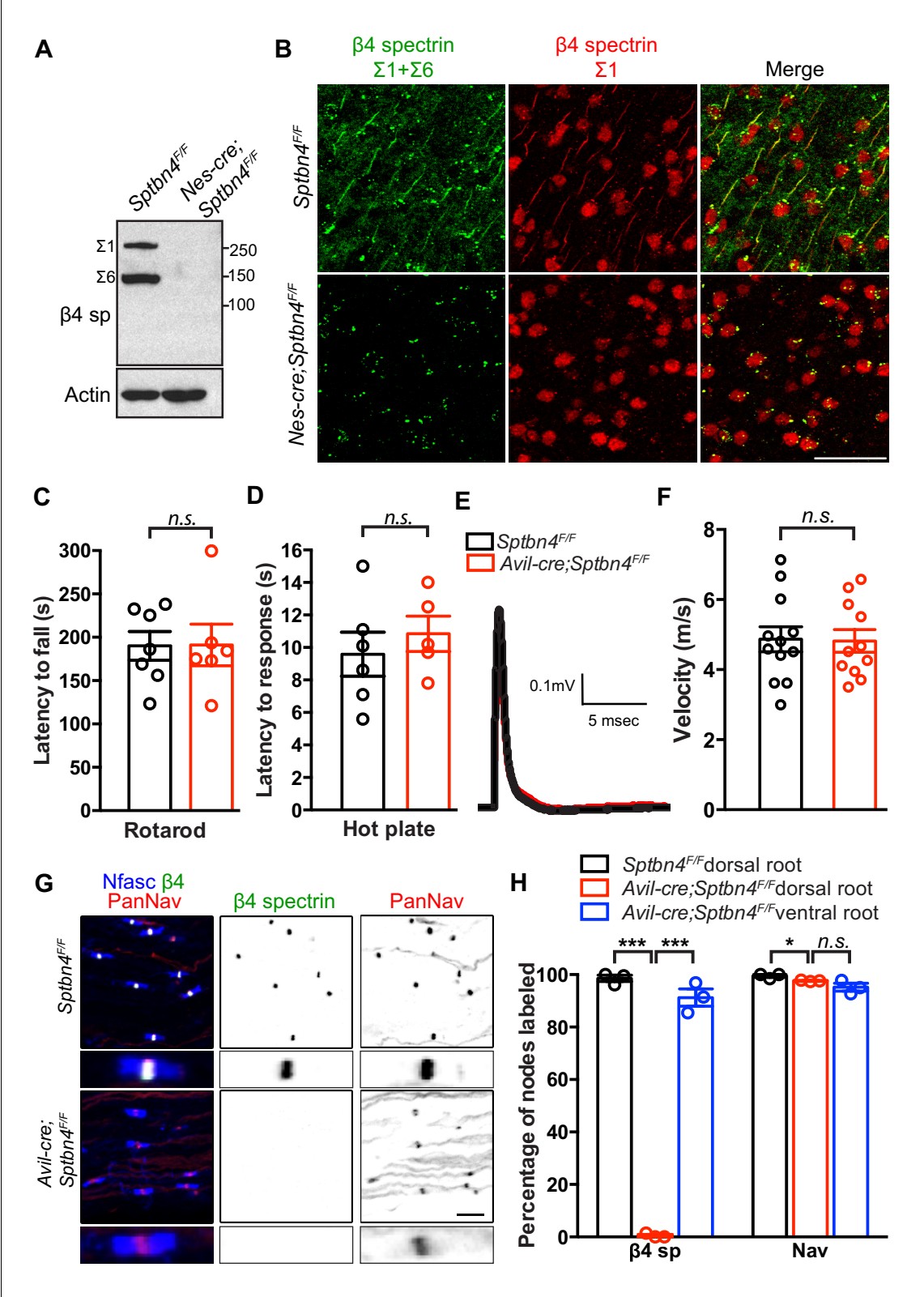

**Figure 1.** Mice lacking β4 spectrin in PNS sensory neurons have normal behaviors, action potential conduction, and Nav channel clustering at nodes or Ranvier. (**A**) Immunoblotting of brain homogenates from 3 month-old *Sptbn4^{F/F}* and *Nes-cre;Sptbn4^{F/F}* mice using antibodies against the C-terminal SD domain of β4 spectrin and actin. (**B**) Immunostaining of cortical brain sections from 3 month-old *Sptbn4^{F/F}* and *Nes-cre;Sptbn4^{F/F}* mice using antibodies against the C-terminal SD domain (green) and N-terminal domain (red) of β4 spectrin. Scale bar, 50 μm. (**C**) Accelerating rotarod test performed on 6

*Figure 1 continued on next page*

*Figure 1 continued*

week-old mice. *Sptbn4$^{F/F}$*, N = 7; *Avil-cre;Sptbn4$^{F/F}$*, N = 6. Data are mean ± SEM, p=0.9708. (**D**) Hot plate test performed on 6 week-old mice. *Sptbn4$^{F/F}$*, N = 6; *Avil-cre;Sptbn4$^{F/F}$*, N = 5. Data are mean ± SEM, p=0.5003. (**E**) Representative compound action potentials recorded from 5 week-old *Sptbn4$^{F/F}$* (black) and *Avil-cre;Sptbn4$^{F/F}$* (red) dorsal roots. (**F**) Conduction velocities recorded from 5 week-old mice. *Sptbn4$^{F/F}$*, N = 3 mice, 12 dorsal roots. *Avil-cre;Sptbn4$^{F/F}$*, N = 3 mice, 11 dorsal roots. Data are mean ± SEM. p=0.9202. (**G**) Immunostaining of dorsal roots from 1.5 month-old *Sptbn4$^{F/F}$* and *Avil-cre;Sptbn4$^{F/F}$* mice using antibodies against pan-neurofascin (blue), β4 spectrin SD-domain (green), and pan-Nav channels (red). Scale bar, 10 µm. (**H**) Quantification of the percentage of dorsal root nodes labeled for β4 spectrin and Nav channels in the indicated tissues and genotypes. N = 3 animals per tissue per genotype, with at least 80 nodes counted per data point. Data are mean ± SEM. For β4 spectrin labeling, ***p=1.97595E-07 between *Sptbn4$^{F/F}$* and *Avil-cre;Sptbn4$^{F/F}$* dorsal roots, or ***p=1.07844E-05 between *Avil-cre;Sptbn4$^{F/F}$* dorsal roots and ventral roots, respectively. For Nav labeling, *p=0.0246 between *Sptbn4$^{F/F}$* and *Avil-cre;Sptbn4$^{F/F}$* dorsal roots, or p=0.1783 between *Avil-cre; Sptbn4$^{F/F}$* dorsal roots and ventral roots, respectively.

The online version of this article includes the following source data for figure 1:

**Source data 1.** Original immunoblotting image for *Figure 1A* and statistical summary for *Figure 1C,D,F,H*.

mice, we measured the latency to fall on an accelerating rotarod and the latency to a response using the hot plate test. We found no significant difference between 1.5 month-old *Sptbn4$^{F/F}$* and *Avil-Cre;Sptbn4$^{F/F}$* groups (*Figure 1C,D*). To determine whether the electrophysiological properties of sensory roots are impaired in *Avil-Cre;Sptbn4$^{F/F}$* mice, we measured compound action potentials (CAPs) in dorsal roots. We found no difference in CAP shape or conduction velocity (*Figure 1E,F*). These results show that sensory neuron function and action potential propagation are intact in *Avil-Cre;Sptbn4$^{F/F}$* mice, and that β4 spectrin is not required for sensory axon function.

To determine whether nodal Nav channel clustering is affected by loss of β4 spectrin, we immunostained dorsal and ventral roots of 1.5 month-old *Sptbn4$^{F/F}$* and *Avil-Cre;Sptbn4$^{F/F}$* mice. We found colocalized β4 spectrin and Nav channels at nodes of Ranvier in dorsal roots of *Sptbn4$^{F/F}$* and ventral roots of *Avil-Cre;Sptbn4$^{F/F}$* mice (*Figure 1G,H*). Although we confirmed loss of β4 spectrin from dorsal root nodes in *Avil-Cre;Sptbn4$^{F/F}$* mice, Nav channels were still enriched at nodes (*Figure 1G,H*); even at 6 months of age we observed no difference in the percentage of nodes with Nav channels (not shown). We also measured node length and found mean node lengths of 1.38 ± 0.05 µm and 1.55 ± 0.12 µm (± SEM, p=0.28) in *Sptbn4$^{F/F}$* and *Avil-Cre;Sptbn4$^{F/F}$* mice, respectively. These results show that β4 spectrin is dispensable for nodal Nav channel clustering.

## β1 spectrin/AnkyrinR comprise a secondary mechanism for Nav channel clustering at nodes

We previously showed that AnkyrinR (AnkR) and β1 spectrin can rescue nodal Nav channel clustering in sensory neuron specific AnkG knockout mice (*Ho et al., 2014*). We also showed that β1 spectrin and AnkR were found at nodes of Ranvier in mutant *Sptbn4$^{qv3j/qv3j}$* mice (*Ho et al., 2014*). However, these mutant *Sptbn4$^{qv3j/qv3j}$* mice express a truncated version of β4 spectrin in all neuronal tissues rather than being full knockouts (*Yang et al., 2004*). To rule out dominant negative effects of the truncated protein in the *Sptbn4$^{qv3j/qv3j}$* mice without the confound of disrupted AIS, we examined the potential compensation by β1 spectrin and AnkR at nodes in *Avil-Cre;Sptbn4$^{F/F}$* mice. Immunostaining showed that β1 spectrin was found at all nodes in *Avil-Cre;Sptbn4$^{F/F}$* mice (*Figure 2A,C*). Similarly, we found that nodal AnkR was significantly increased while nodal AnkG decreased in *Avil-Cre;Sptbn4$^{F/F}$* mice (*Figure 2B,C*). These results support the existence of a hierarchy of spectrins and ankyrins at nodes of Ranvier (*Ho et al., 2014*). The primary complex consists of β4 spectrin and AnkG, but when either β4 spectrin or AnkG are absent, a secondary complex consisting of β1 spectrin and AnkR can substitute.

Is β1 spectrin important for the normal assembly of nodes of Ranvier? Or is it dispensable as a secondary, back-up spectrin when β4 spectrin is present? To test this possibility, we generated a conditional null allele of β1 spectrin by flanking exon 2 of the mouse *Sptb* gene with *loxP* sites (*Sptb$^{F/F}$* mice); in the presence of Cre recombinase exon two is excised. To confirm the efficiency of deletion, we first removed β1 spectrin from the central nervous system by crossing *Sptb$^{F/F}$* mice with *Nes-Cre* mice. Immunoblotting of brain homogenate and immunostaining of brain sections showed complete removal of neuronal β1 spectrin in *Nes-Cre;Sptb$^{F/F}$* mice (*Figure 3A,B*). To determine if β1 spectrin is necessary for peripheral sensory neuron function and nodal Nav channel clustering, we generated *Avil-Cre;Sptb$^{F/F}$* mice and performed rotarod, hot plate, and nerve conduction studies.

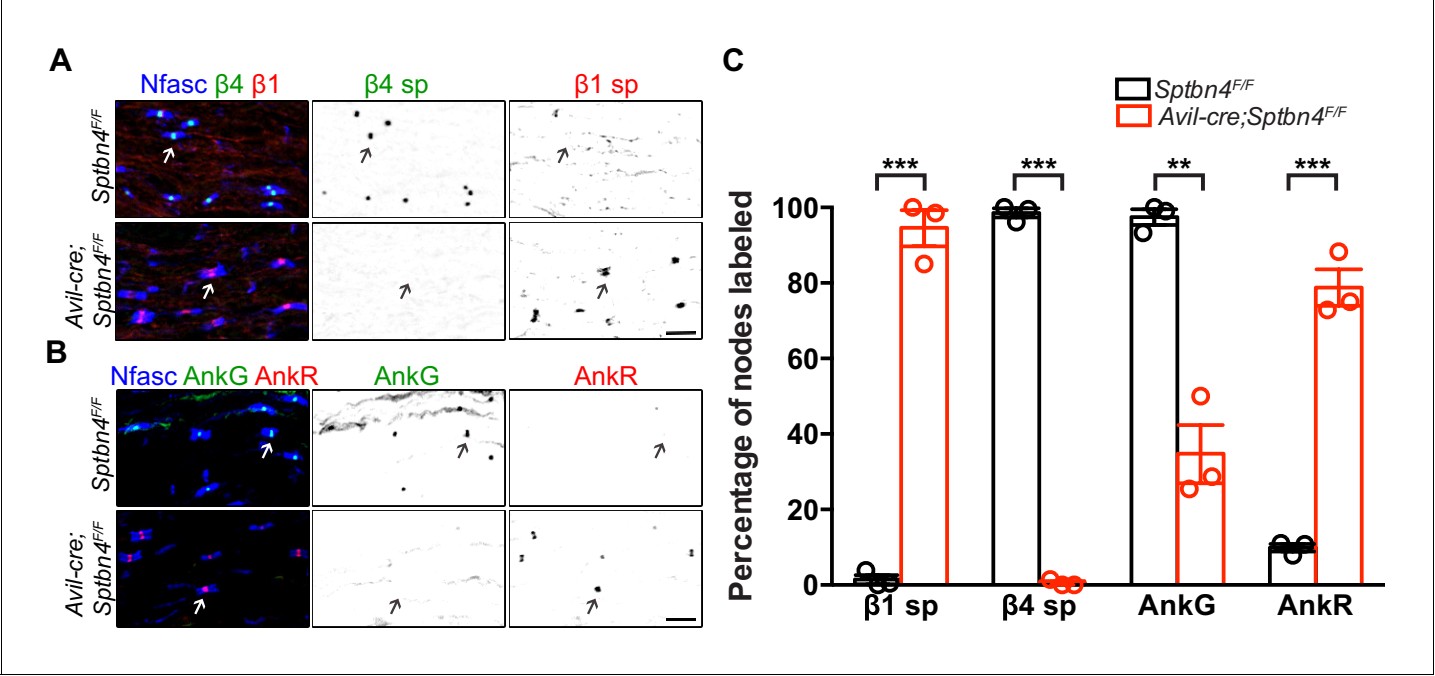

**Figure 2.** β1 spectrin and AnkR rescue Nav channel clustering at nodes in β4 spectrin deficient dorsal roots. (**A**) Immunostaining of dorsal roots from 1.5 month-old *Sptbn4^{F/F}* and *Avil-cre;Sptbn4^{F/F}* mice using antibodies against pan-neurofascin (blue), β4 spectrin SD-domain (green), and β1 spectrin (red). Scale bar, 10 μm. (**B**) Immunostaining of dorsal roots from 1.5 month-old *Sptbn4^{F/F}* and *Avil-cre;Sptbn4^{F/F}* mice using antibodies against pan-neurofascin (blue), AnkG (green), and AnkR (red). Scale bar, 10 μm. (**C**) Quantification of the percentage of dorsal root nodes in *Sptbn4^{F/F}* and *Avil-cre; Sptbn4^{F/F}* mice labeled for β1 spectrin, β4 spectrin, AnkG and AnkR. N = 3 mice per genotype with more than 80 nodes counted per data point. Data are mean ± SEM. β1 spectrin, ***p=4.62544E-05; β4 spectrin, ***p=1.97595E-07; AnkG, **p=0.0014; AnkR, ***p=0.0002.

The online version of this article includes the following source data for figure 2:

**Source data 1.** Statistical summary for *Figure 2C*.

Behavioral and electrophysiological results showed no significant differences between control and *Avil-Cre;Sptb^{F/F}* mice (*Figure 3C–F*). Furthermore, immunostaining of sensory roots from 1.5 month-old *Avil-Cre;Sptb^{F/F}* mice showed that β4 spectrin and Nav channels are normally clustered at nodes (*Figure 3G–H*). Together, these results support the notion of a hierarchy of nodal spectrin function and show that β1 spectrin does not normally contribute to Nav channel clustering or peripheral sensory neuron function.

## Nodal spectrins are required to maintain nodal nav channel clusters

What is the function of the nodal spectrin cytoskeleton? To selectively and simultaneously eliminate β4 and β1 spectrins from nodes without affecting AIS spectrins, we generated *Avil-Cre;Sptb^{F/F};Sptbn4^{F/F}* mice. We found that *Avil-Cre;Sptb^{F/F};Sptbn4^{F/F}* mice had severe defects in proprioception as indicated by hindlimb clasping (*Figure 4A*) and gait analysis (*Figure 4B* and *Video 1*). These impairments prevented *Avil-Cre;Sptb^{F/F};Sptbn4^{F/F}* mice from remaining on the accelerating rotarod (*Figure 4C*). However, consistent with a specific impairment at nodes of Ranvier, nociception as assayed by the tail immersion test, showed no significant difference in the latency to tail flick (*Figure 4D*). Furthermore, we measured compound action potentials in dorsal roots and found both amplitude and velocity were significantly decreased in *Avil-Cre;Sptb^{F/F};Sptbn4^{F/F}* mice (*Figure 4E, F*). Thus, mice lacking both β4 and β1 spectrins have severely compromised myelinated sensory axon function.

Since action potential propagation is impaired in *Avil-Cre;Sptb^{F/F};Sptbn4^{F/F}* mice, we determined whether nodal Nav channels are clustered and if the overall structure of nodes of Ranvier was still intact. Surprisingly, in the absence of β4 and β1 spectrins we found Nav channels were appropriately clustered at nodes at postnatal day 8 (*Figure 5A,B*); we found no difference in the total number of nodes between *Sptb^{F/F};Sptbn4^{F/F}* and *Avil-Cre;Sptb^{F/F};Sptbn4^{F/F}* mice at 1 or 6 months of age

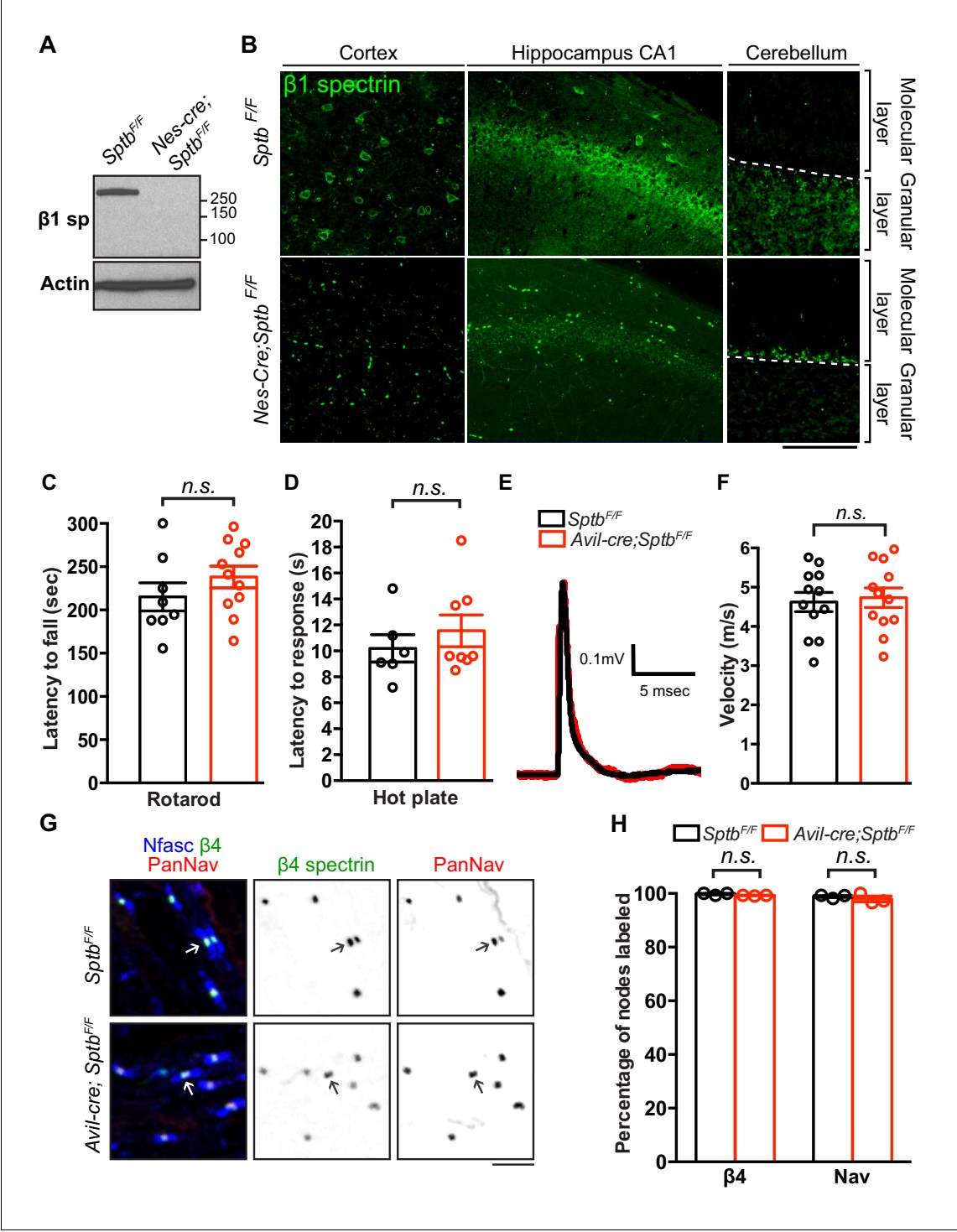

**Figure 3.** Mice lacking β1 spectrin in PNS sensory neurons have normal behaviors, action potential transmission, and Nav channel clustering at nodes. (A) Immunoblotting of brain homogenates from 3 month-old $Sptb^{F/F}$ and $Nes-cre;Sptb^{F/F}$ mice using antibodies against β1 spectrin and actin. (B) Immunostaining of cortex, hippocampal CA1 region and cerebellum from 3 month-old $Sptb^{F/F}$ and $Nes-cre;Sptb^{F/F}$ mice using antibodies against β1 spectrin (green). Scale bar, 50 μm. (C) Accelerating rotarod test performed on 6 week-old mice. $Sptb^{F/F}$, N = 8; $Avil-cre;Sptb^{F/F}$, N = 11. Data are mean ± SEM, p=0.2706. (D) Hot plate test performed on 6 week-old mice. $Sptb^{F/F}$, N = 6; $Avil-cre;Sptb^{F/F}$, N = 8. Data are mean ± SEM, p=0.4403. (E) Representative CAPs recorded from 5 week-old $Sptb^{F/F}$ (black) and $Avil-cre;Sptb^{F/F}$ (red) dorsal roots. (F) Conduction velocities recorded from 5 week-old mice. $Sptb^{F/F}$, N = 3 mice, 12 dorsal roots. $Avil-cre;Sptb^{F/F}$, N = 3 mice, 12 dorsal roots. Data are mean ± SEM. p=0.7562. (G) Immunostaining of dorsal roots from 6-week-old $Sptb^{F/F}$ and $Avil-cre;Sptb^{F/F}$ mice using antibodies against pan-neurofascin (blue), β4 spectrin SD-domain (green), and
*Figure 3 continued on next page*

*Figure 3 continued*

pan-Nav channels (red). Scale bar, 10 μm. (**H**) Quantification of the percentage of dorsal root nodes labeled for β4 spectrin and Nav channels in $Sptb^{F/F}$ and $Avil$-$cre;Sptb^{F/F}$ mice. N = 3 animals per genotype, with at least 80 nodes counted per animal. Data are mean ± SEM.

The online version of this article includes the following source data for figure 3:

**Source data 1.** Original immunoblotting image for *Figure 3A* and statistical summary for *Figure 3C,D,F,H*.

(*Figure 5—figure supplement 1*). However, compared to $Sptb^{F/F};Sptbn4^{F/F}$ mice, the number of nodes with nodal Nav channels in $Avil$-$Cre;Sptb^{F/F};Sptbn4^{F/F}$ mice progressively decreased with increasing age such that by 6 months of age only ~40% of nodes had Nav channels (*Figure 5A,B*). Immunostaining also revealed that nodes in $Avil$-$Cre;Sptb^{F/F};Sptbn4^{F/F}$ mice had both AnkR and AnkG (*Figure 5C*), since both can bind to NF186 (*Ho et al., 2014*), but the number of nodes with ankyrins decreased with increasing age. Despite the significant loss of nodal Nav channels in $Avil$-$Cre;Sptb^{F/F};Sptbn4^{F/F}$ mice, the absence of β4 and β1 spectrin did not affect paranodal axon-glia interactions (as indicated by Caspr immunostaining) or juxtaparanodal domains (as indicated by Kv1.2 immunostaining) in 6 month-old $Avil$-$Cre;Sptb^{F/F};Sptbn4^{F/F}$ mice (*Figure 5D*). Together, these

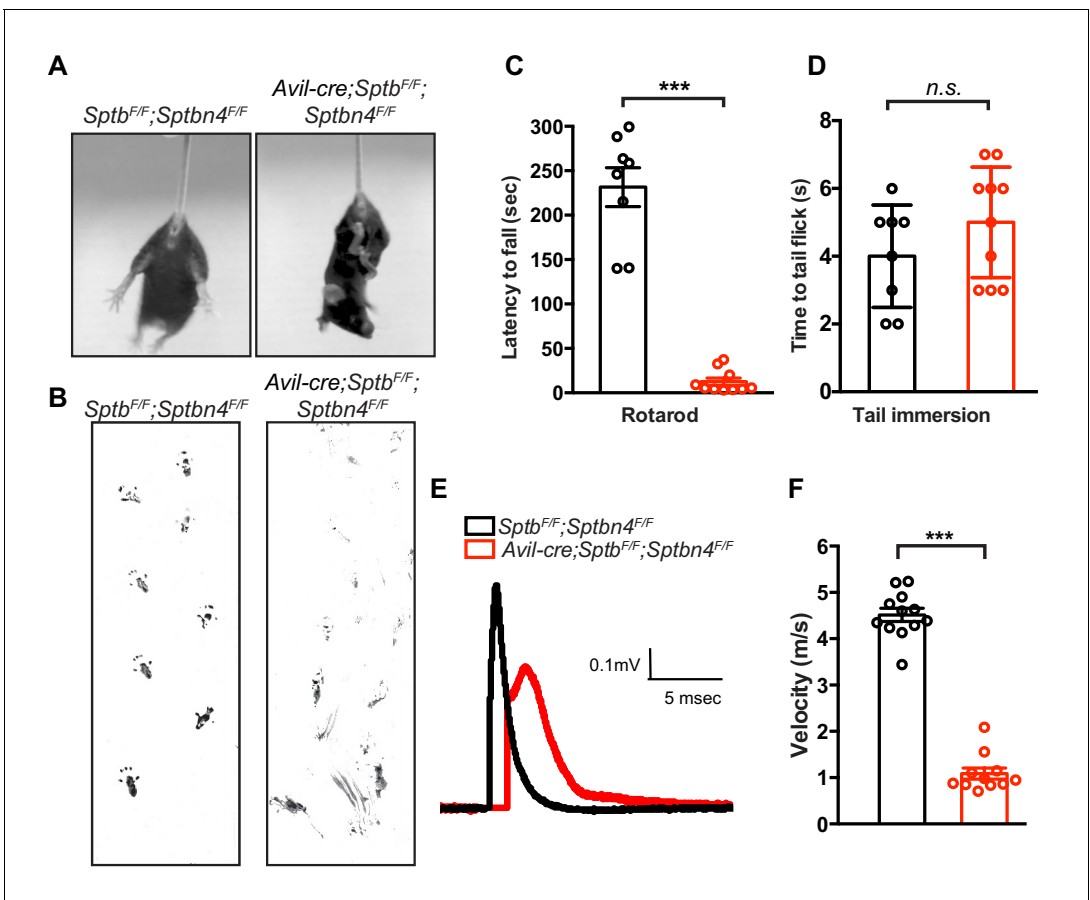

**Figure 4.** Mice lacking both β1 and β4 spectrin in PNS sensory neurons have motor dysfunction and impaired action potential conduction. (**A**) $Avil$-$cre;Sptb^{F/F};Sptbn4^{F/F}$ mice have a hindlimb clasping reflex. (**B**) Footprint assay in 6 week-old $Sptb^{F/F};Sptbn4^{F/F}$ and $Avil$-$cre;Sptb^{F/F};Sptbn4^{F/F}$ mice. (**C**) Accelerating rotarod test performed on 6 week-old mice. $Sptb^{F/F};Sptbn4^{F/F}$, N = 8; $Avil$-$cre; Sptb^{F/F};Sptbn4^{F/F}$, N = 10. Data are mean ± SEM, ***p=7.17995E-09. (**D**) Tail immersion test performed on 6 week-old mice. $Sptb^{F/F}$, N = 8; $Avil$-$cre;Sptb^{F/F};Sptbn4^{F/F}$, N = 10. Data are mean ± SEM, p=0.2011. (**E**) Representative CAPs recorded from 5 week-old $Sptb^{F/F};Sptbn4^{F/F}$ (black) and $Avil$-$cre;Sptb^{F/F};Sptbn4^{F/F}$ (red) dorsal roots. (**F**) Conduction velocities recorded from 5 week-old mice. $Sptb^{F/F};Sptbn4^{F/F}$, N = 3 mice, 12 dorsal roots. $Avil$-$cre;Sptb^{F/F};Sptbn4^{F/F}$, N = 3 mice, 11 dorsal roots. Data are mean ± SEM. ***p=2.76334E-14.

The online version of this article includes the following source data for figure 4:

**Source data 1.** Statistical summary for *Figure 4C,D,F*.

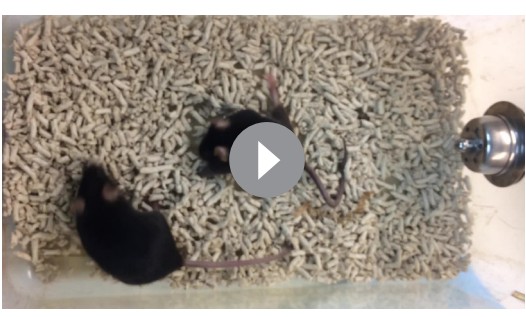

**Video 1.** 9 month-old WT and *Avil-cre;Sptb^{F/F};Sptbn4^{F/F}* mice. Mice lacking both β1 and β4 spectrin in sensory neurons are ataxic.
https://elifesciences.org/articles/52378#video1

results show that the nodal spectrin cytoskeleton does not contribute to the initial clustering of Nav channels, but instead is required to maintain Nav channels and ankyrins at nodes.

To determine whether other axonal β spectrins (*Stankewich et al., 2010*; *Zhang et al., 2013*) can partially compensate for loss of β1 and β4 spectrins, we immunostained teased dorsal roots of *Avil-Cre;Sptb^{F/F};Sptbn4^{F/F}* mice using antibodies against β2 and β3 spectrin. We did not detect either of these β spectrins at nodes (*Figure 5E,F*). Thus, despite expression of β2 and β3 spectrin in axons, only β1 spectrin or β4 spectrin can function at nodes to maintain Nav channels.

## The nodal spectrin cytoskeleton is critical for axon health

Loss of α2 spectrin causes axon degeneration (*Huang et al., 2017a*; *Huang et al., 2017b*). However, loss of internodal β2 spectrin does not cause axon degeneration in PNS sensory axons (*Zhang et al., 2013*), suggesting that loss of the nodal cytoskeleton alone may be sufficient to drive axon injury and degeneration in sensory neurons lacking an AIS. To test this possibility, we collected dorsal root ganglia (DRG) from wild-type, *Avil-Cre;Sptb^{F/F}*, *Avil-Cre;Sptbn4^{F/F}* and *Avil-Cre;Sptb^{F/F};Sptbn4^{F/F}* mice at 1, 6, and 9 months of age. As an indicator of the axon injury response, we examined ATF3 expression in the DRG by immunostaining (*Tsujino et al., 2000*). We found no significant difference among all genotypes in 1 month-old mice (*Figure 6A,B*). However, the percentage of ATF3 positive DRG neurons was dramatically and significantly increased in neurons lacking both β1 and β4 spectrin in 6 and 9+ months-old mice (*Figure 6A,B*). However, we did not detect any ATF3 labeling of DRG neurons in mice lacking only a single β spectrin. Consistent with the behavioral phenotypes, we found that large diameter neurons were more likely to be ATF3+ (*Figure 6C*).

To further investigate how loss of nodal β spectrins affects axons, we examined axon ultrastructure. Compared to control mice, *Avil-Cre;Sptb^{F/F};Sptbn4^{F/F}* axons were deformed and irregular (*Figure 6D,E*). Measurement of the g-ratio suggested that *Avil-Cre;Sptb^{F/F};Sptbn4^{F/F}* axons are more myelinated than axons from *Sptb^{F/F};Sptbn4^{F/F}* mice (*Figure 6F,G*). Furthermore, quantification of axon diameter revealed the mean diameter of axons lacking nodal β spectrins is significantly less than that for control axons (*Figure 6H,I*). In summary, loss of nodal β spectrins alone is sufficient to disrupt axon structure and induce an axon injury response.

## Discussion

Clustering of Nav channels at nodes of Ranvier is essential for efficient action potential propagation and has been proposed as a key event in the evolution of complex nervous systems (*Hill et al., 2008*). During development, two overlapping glia-dependent mechanisms rapidly cluster ankyrins and Nav channels (*Amor et al., 2017*; *Feinberg et al., 2010*; *Susuki et al., 2013*). After their initial recruitment, nodal Nav channel clusters must also be maintained throughout the lifetime of the animal. Thus, robust mechanisms must exist to stabilize and maintain nodal protein complexes – for years and even decades. The nodal spectrin cytoskeleton, normally consisting of tetramers of two β4 and two α2 spectrin subunits, has been proposed to perform this function. Indeed, loss of α2 spectrin causes axon degeneration (*Huang and Rasband, 2018*; *Huang et al., 2017b*). However, since α2 spectrin is located throughout the neuron, and β4 spectrin is highly enriched at the AIS, it is difficult to discern their functions at the AIS (α2 and β4 spectrin) and internodes (α2 spectrin) from their functions at nodes. Further complicating matters is the observation that in *Sptbn4^{qv3j/qv3J}* (β4 spectrin mutant) mice, β1 spectrin can compensate for the progressive loss of β4 spectrin from nodes of Ranvier (*Ho et al., 2014*). In addition, mice and humans with pathogenic *Sptbn4* variants have intact sensory functions, but impaired motor function and axon degeneration, suggesting that β4 spectrin is dispensable in some contexts, but necessary in others (*Wang et al., 2018*). By analyzing sensory

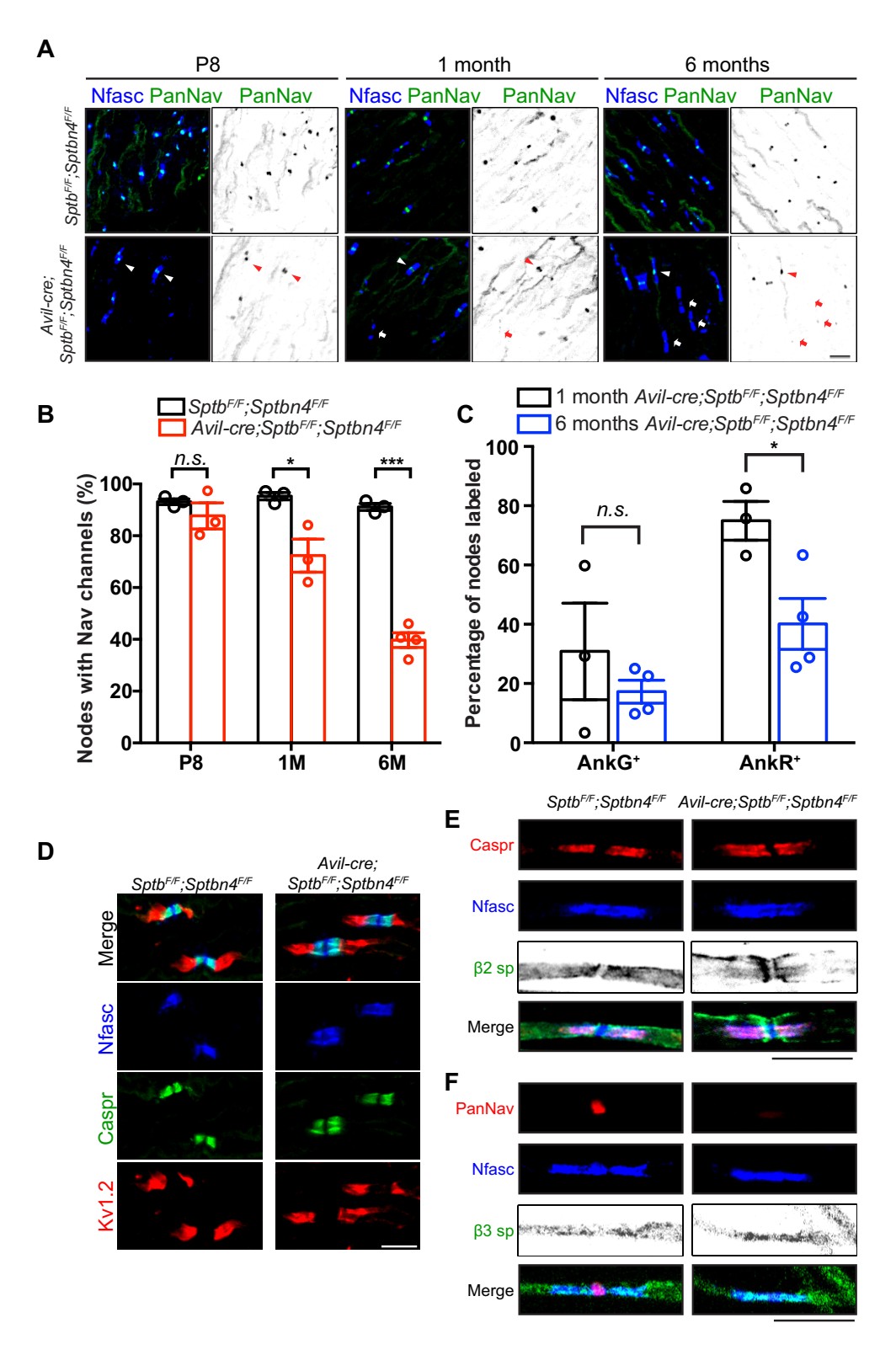

**Figure 5.** Mice lacking both β1 and β4 spectrin in PNS sensory neurons progressively lose their nodal Nav channels. (A) Immunostaining of dorsal roots from P8, 1 month-old and 6 month-old $Sptb^{F/F};Sptbn4^{F/F}$ and $Avil\text{-}cre;Sptb^{F/F};Sptbn4^{F/F}$ mice using antibodies against pan-neurofascin (blue) and panNav channels (green). Arrowheads point to intact nodal Nav clusters, whereas arrows point to nodes devoid of Nav channels. Scale bar, 10 μm. (B) Quantification of the percentage of dorsal root nodes labeled for Nav channels in $Sptb^{F/F};Sptbn4^{F/F}$ and $Avil\text{-}cre;Sptb^{F/F};Sptbn4^{F/F}$ mice at the

*Figure 5 continued on next page*

*Figure 5 continued*

indicated ages. N = 3 animals per genotype per time point, except N = 4 in 6 month-old *Avil-cre;Sptb^{F/F};Sptbn4^{F/F}*. Data are mean ± SEM. P8: p=0.3508; 1M: *p=0.0249; 6M:***p=2.8444E-05. (C) The percentage of nodes labeled for AnkG or AnkR at 1 month or 6 months of age in *Avil-cre; Sptb^{F/F};Sptbn4^{F/F}* mice. N = 3 or four animals per genotype per timepoint, with a minimum of 80 nodes analyzed per animal. Data are mean ± SEM, *p=0.0296. (D) Immunostaining of dorsal roots from 6 month-old *Sptb^{F/F};Sptbn4^{F/F}* and *Avil-cre;Sptb^{F/F};Sptbn4^{F/F}* mice using antibodies against pan-neurofascin (blue), caspr (green), and Kv1.2 (red). Scale bar, 10 µm. (E) Immunostaining of teased dorsal roots from 9 month-old *Sptb^{F/F};Sptbn4^{F/F}* and *Avil-cre;Sptb^{F/F};Sptbn4^{F/F}* mice using antibodies against pan-neurofascin (blue), β2 spectrin (green), and caspr (red). Scale bar, 10 µm. (F) Immunostaining of teased dorsal roots from 9 month-old *Sptb^{F/F};Sptbn4^{F/F}* and *Avil-cre;Sptb^{F/F};Sptbn4^{F/F}* mice using antibodies against pan-neurofascin (blue), β3 spectrin (green), and panNav channels (red). Scale bar, 10 µm.

The online version of this article includes the following source data and figure supplement(s) for figure 5:

**Source data 1.** Statistical summary for *Figure 5B,C*.
**Figure supplement 1.** Quantification of nodes per field of view (FOV) by paranodal neurofascin immunostaining in *Sptb^{F/F};Sptbn4^{F/F}* and *Avil-cre; Sptb^{F/F};Sptbn4^{F/F}* mice dorsal root at the indicated ages.
**Figure supplement 1—source data 1.** Statistical summary for *Figure 5—figure supplement 1*.

neuron specific conditional knockouts for β1, β4, and β1/ β4 spectrins we resolved many of these confounds and were able to define the function of the nodal spectrin cytoskeleton. For example, by removing nodal β spectrins specifically from sensory neurons that lack an AIS (*Gumy et al., 2017*), we uncoupled their role at the AIS from their role at nodes.

Our experiments confirmed the existence of a hierarchy of nodal β spectrins. This hierarchy was first proposed in experiments by *Ho et al. (2014)* where we showed that an ankyrin/spectrin complex consisting of AnkR/β1 spectrin can substitute for AnkG/β4 spectrin. Because of this compensation, to define the role of nodal ankyrins we had to remove both AnkR and AnkG. Loss of both ankyrins completely blocked nodal Nav channel clustering during node assembly. However, a similar experiment to test the requirement of nodal β spectrins was not possible since conditional alleles for *Sptb* and *Sptbn4* were not available at that time. Instead, we used *Sptbn4^{qv3j/qv3J}* mice that express a truncated version of β4 spectrin throughout the nervous system and that is not stable at nodes or AIS, to show that β1 spectrin can be found at nodes lacking β4 spectrin (*Ho et al., 2014*). In the present study we significantly improved our experimental approach by using β4 and β1 spectrin conditional knockout mice to finally determine the nodal function of β spectrins. We confirmed the main spectrin found at mature nodes of Ranvier is β4 spectrin, while β1 spectrin can substitute when β4 spectrin is missing (*Figure 7*). Removing a single nodal β spectrin revealed that neither β1 nor β4 spectrin is required for assembly or maintenance of nodal Nav channels. Furthermore, removing both β1 and β4 spectrin from nodes simultaneously showed that in contrast to ankyrins, a nodal spectrin cytoskeleton is not required for the initial clustering of Nav channels (*Figure 7*); this observation is consistent with the fact that β4 spectrin is recruited to nodes of Ranvier through binding to AnkG (*Jenkins et al., 2015*; *Yang et al., 2007*). Instead, we find that nodal β spectrins are necessary to maintain nodal Nav channel clusters (*Figure 7*). Furthermore, although a β2 spectrin-dependent paranodal barrier can cluster nodal proteins during development (*Amor et al., 2017*), our results show that in the absence of nodal spectrins (α2/β1 and α2/β4), paranodal spectrins (α2/β2) are not sufficient to maintain Nav channel protein complexes; in addition, we did not see any compensation by other spectrins at nodes. Previous studies showed that PNS nodal extracellular matrix molecules including gliomedin and NrCAM help to maintain proper nodal Nav channel densities (*Amor et al., 2014*). Our experiments extend these observations and show that both extracellular and intracellular interactions preserve nodal Nav channel clusters.

Mice and humans with pathogenic variants of β4 spectrin have been investigated extensively. However, this is the first report examining loss of β1 spectrin in neurons. Importantly, mice lacking β1 spectrin in sensory neurons had no phenotype, suggesting that despite its robust expression (*Ho et al., 2014*), β1 spectrin is not required in sensory neurons. Thus, the normal function of β1 spectrin in axons remains unknown. Future studies examining the role of β1 spectrin in other neuron types may reveal additional neuronal functions for this β spectrin beyond its canonical functions in red blood cells. However, our studies do not explain why humans with pathogenic variants of β4 spectrin have widespread nervous system dysfunction except in sensory neurons (*Wang et al., 2018*). We propose that future studies using *Sptb^{F/F}* and *Sptbn4^{F/F}* mice together with other mouse lines that express Cre recombinase in neurons with an AIS (e.g. motor neurons) may help to reveal

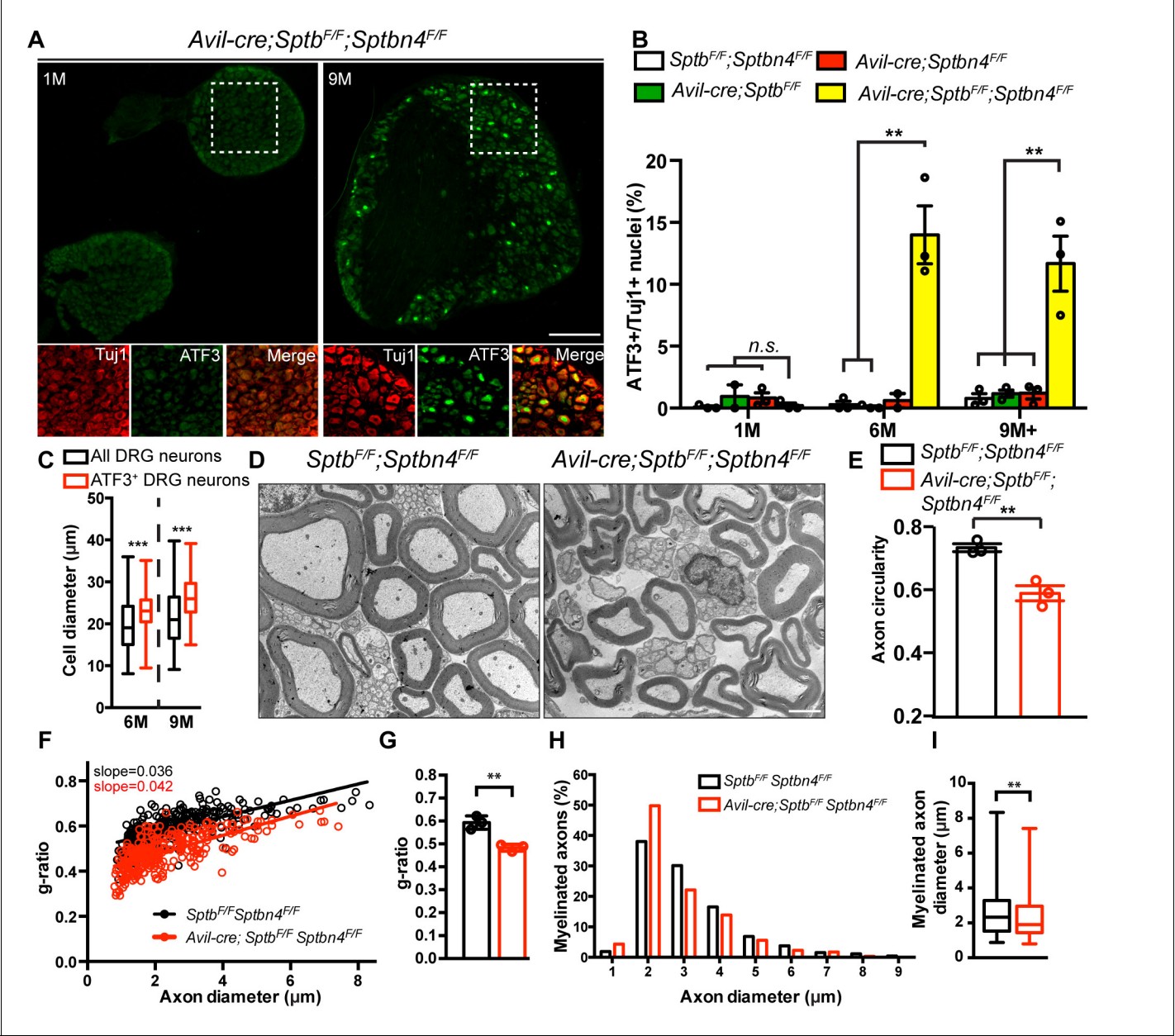

**Figure 6.** PNS sensory axons lacking nodal spectrins have an axon injury response, deformation, and hypermyelination. (**A**) Immunostaining of DRG from 1 and 9 month-old *Avil-cre;Sptb^F/F^;Sptbn4^F/F^* mice using antibodies against ATF3 (green) and the axonal marker Tuj1 (red). Scale bar, 200 μm. (**B**) Quantification of the percentage of Tuj1-positive DRG neurons labeled for ATF3 in *Sptb^F/F^;Sptbn4^F/F^*, *Avil-cre;Sptb^F/F^*, *Avil-cre;Sptbn4^F/F^* and *Avil-cre; Sptb^F/F^;Sptbn4^F/F^* mice at the indicated ages. N = 3 mice per genotype per age, except N = 2 for 1 month-old *Avil-cre;Sptb^F/F^*, and N = 2 for 6 month-old *Avil-cre;Sptbn4^F/F^* mice. For 1 month-old: *Sptb^F/F^;Sptbn4^F/F^ vs. Avil-cre;Sptb^F/F^;Sptbn4^F/F^*, p=0.6321; *Avil-cre;Sptbn4^F/F^ vs. Avil-cre;Sptb^F/F^;Sptbn4^F/F^*, p=0.2712. For 6 month-old: *Sptb^F/F^;Sptbn4^F/F^ vs. Avil-cre;Sptb^F/F^;Sptbn4^F/F^*, \*\*p=0.0044; *Avil-cre;Sptb^F/F^ vs. Avil-cre;Sptb^F/F^;Sptbn4^F/F^*, \*\*p=0.0041; For 9+ months old: *Sptb^F/F^;Sptbn4^F/F^ vs. Avil-cre;Sptb^F/F^;Sptbn4^F/F^*, \*\*p=0.0085; *Avil-cre;Sptb^F/F^ vs. Avil-cre;Sptb^F/F^;Sptbn4^F/F^*, \*\*p=0.0094; *Avil-cre; Sptbn4^F/F^ vs. Avil-cre;Sptb^F/F^;Sptbn4^F/F^*, \*\*p=0.0099. Data are mean ± SEM. (**C**) Quantification of total and ATF3^+^ DRG cell body diameters in *Avil-cre; Sptb^F/F^;Sptbn4^F/F^* mice at the indicated ages. N = 3 animals per time point. Data are shown in a box-and-whisker plot (median: a line across 25th and 75th percentiles: lower and upper box edges, respectively; minimum and maximum: the values below and above the box, respectively). Total number of DRG neurons counted are 1078 at 6M, and 1005 at 9M. At 6M: \*\*\*p=6.61E-07; 9M: \*\*\*p=8.06E-10. (**D**) TEM images of 9 month-old dorsal root cross sections from *Sptb^F/F^;Sptbn4^F/F^* and *Avil-cre;Sptb^F/F^;Sptbn4^F/F^* mice. Scale bar, 2 μm. (**E**) Quantification of axon circularity in 9 month-old *Sptb^F/F^; Sptbn4^F/F^* and *Avil-cre;Sptb^F/F^;Sptbn4^F/F^* mice. N = 3 mice per genotype. *Sptb^F/F^;Sptbn4^F/F^*, n = 533 axons; *Avil-cre;Sptb^F/F^;Sptbn4^F/F^*, n = 873 axons. \*\*p=0.0056. Data are mean ± SEM. (**F**) Scatter plot of g-ratio versus axon diameter from dorsal roots of 9 month-old *Sptb^F/F^;Sptbn4^F/F^* and *Avil-cre; Sptb^F/F^;Sptbn4^F/F^* mice. *Sptb^F/F^Sptbn4^F/F^*, n = 266 axons; *Avil-cre; Sptb^F/F^;Sptbn4^F/F^*, n = 303 axons pooled from 3 mice of each genotype. (**G**) Quantification of g-ratio of dorsal roots from 9 month-old *Sptb^F/F^;Sptbn4^F/F^* and *Avil-cre;Sptb^F/F^;Sptbn4^F/F^* mice. N = 3 mice per genotype. \*\*p=0.0045.

*Figure 6 continued on next page*

*Figure 6 continued*

Data are mean ± SEM. (H) Percentage of dorsal root axon diameter in 9 month-old *Sptb^{F/F};Sptbn4^{F/F}* and *Avil-cre; Sptb^{F/F};Sptbn4^{F/F}* mice. Axons are shown in 1 μm bins. (I) Quantification of axon diameter of dorsal root from 9-month-old *Sptb^{F/F};Sptbn4^{F/F}* and *Avil-cre; Sptb^{F/F};Sptbn4^{F/F}* mice. Data are shown in a box-and-whisker plot (median: a line across the box; 25th and 75th percentiles: lower and upper box edges, respectively; minimum and maximum: the values below and above the box, respectively). **p=0.0023. In (H) and (I), *Sptb^{F/F};Sptbn4^{F/F}*, n = 266 axons; *Avil-cre; Sptb^{F/F};Sptbn4^{F/F}*, n = 303 axons pooled from 3 mice of each genotype.

The online version of this article includes the following source data for figure 6:

**Source data 1.** Statistical summary and individual data for *Figure 6B,C,E,F,G,H,I*.

the cause of the pathology. We speculate that β1 spectrin cannot compensate for β4 spectrin at AIS. This would destabilize AIS proteins, resulting in the loss of AIS Nav channels and AnkG. Loss of AnkG disrupts neuronal polarity and causes axon degeneration (*Hedstrom et al., 2008*; *Sobotzik et al., 2009*; *Zhou et al., 1998*).

Long peripheral sensory and motor axons constantly experience mechanical stresses including tension and torque due to the movement of joints and limbs. The periodic spectrin-actin cytoskeleton has been proposed to resist these stresses and help to maintain the structural integrity of axons (*Krieg et al., 2017*; *Xu et al., 2013*). Myelinated axons may have an additional layer of protection in the form of a myelin sheath that could also support axons and help withstand mechanical forces and strains. However, the sheath is interrupted at regularly spaced intervals at the nodes of Ranvier, which could make nodes particularly vulnerable to injury. Although spectrins are found throughout the axon in regions covered by the myelin sheath, it seems a unique spectrin cytoskeleton is located at the nodes of Ranvier. We found that specifically and simultaneously ablating nodal β1 and β4 spectrin is sufficient to induce an axon injury response even though the remaining internodal and paranodal spectrin cytoskeletons remain intact. In contrast, mice lacking paranodal and internodal β2 spectrin in sensory neurons showed no axon degeneration (*Zhang et al., 2013*). On the other hand, mice lacking α2 spectrin in sensory neurons had a much stronger phenotype; an axon injury

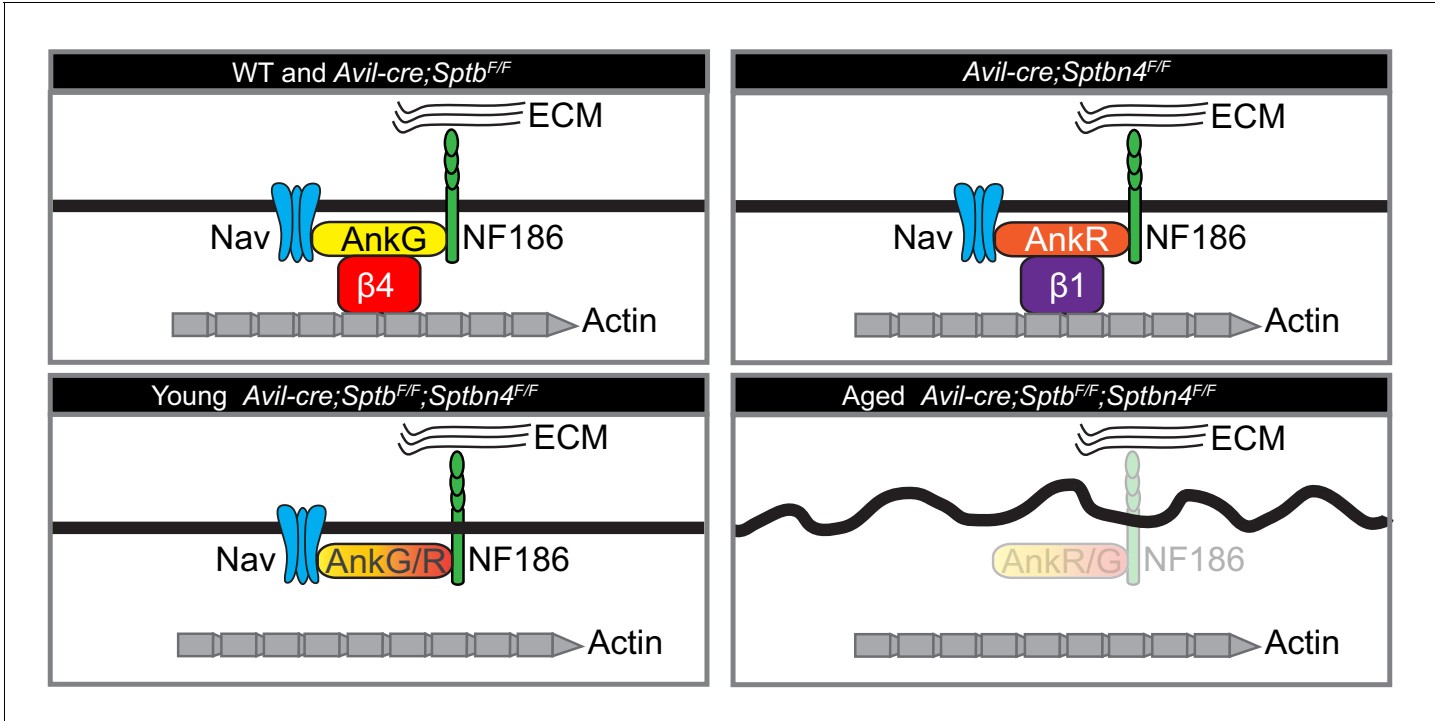

**Figure 7.** Cartoon illustrating the phenotypes of the mice analyzed here. Mice lacking β1 spectrin have normal nodes of Ranvier. Mice lacking β4 spectrin have normal nodes of Ranvier due to compensation by a β1 spectrin/AnkR protein complex. Mice lacking all nodal β spectrins cluster Nav channels during early development. However, with increasing age these mice lose nodal Nav channels, Ankyrins, and NF186, and show an axon injury response including altered membrane organization.

response was detectable in large diameter sensory neurons by two weeks of age and robust axon degeneration was seen by electron microscopy as early as one week after birth (*Huang et al., 2017b*). Mice lacking both β1 and β4 spectrin in sensory neurons showed a less severe axon injury response that was not detectable until the mice were 6 months old. This injury response was accompanied by membrane disorganization as indicated by altered axon circularity. We speculate that loss of nodal spectrins creates intermittent 'breaks' in the spectrin cytoskeleton which gradually compromises the physical properties of the whole network. Previous studies modeling the properties of a periodic arrangement of spectrins suggested they confer more stiffness to axons than the soma and dendrites, and that axonal spectrins are indeed under higher entropic tension due to its fully-stretched organization with 190 nm spacing (*Zhang et al., 2017*). Scattered breaks in the spectrin network may not allow recovery to the initial 'under-tension' configuration. Thus, axons may progressively lose the ability to resist mechanical stresses and eventually degenerate.

These observations on the role of nodal spectrins and axon integrity may have important implications for some diseases and injuries where nodes of Ranvier are a preferential site of $Ca^{2+}$ entry. Elevated $Ca^{2+}$ activates calpain proteases that efficiently degrade spectrin cytoskeletons (*Ma, 2013*; *Schafer et al., 2009*). Thus, disruption of the nodal cytoskeleton may be one important component of peripheral nerve injury and a potential therapeutic point of intervention. In conclusion, we provide evidence for a hierarchy of β spectrins at nodes of Ranvier that are necessary to maintain nodal Nav channel clustering. Furthermore, we show that loss of nodal β spectrins is sufficient to induce an axon injury response.

# Materials and methods

## Key resources table

| Reagent type (species) or resource | Designation | Source or reference | Identifiers | Additional information |
|---|---|---|---|---|
| Gene (*Mus musculus*) | *Sptb* | https://www.ncbi.nlm.nih.gov/gene/20741 | Gene ID: 20741 | |
| Gene (*M. musculus*) | *Sptbn4* | https://www.ncbi.nlm.nih.gov/gene/80297 | Gene ID: 80297 | |
| Genetic reagent (*M. musculus*) | Advillin-cre | F. Wang (Duke University, Durham, NC) | The Jackson Laboratory (Stock No:032536) | |
| Genetic reagent (*M. musculus*) | *Sptb*$^{flox/flox}$ | This paper | | See Materials and methods, Section Animals |
| Genetic reagent (*M. musculus*) | *Sptbn4*$^{flox/flox}$ | PMID: 30226828 | | |
| Antibody | Anti-Ankyrin G (Mouse monoclonal) | Neuromab | Clone: N106/36; RRID: AB_10673030 | IF (1:500) |
| Antibody | Anti-PanNav (Mouse monoclonal) | Neuromab | Clone: N419/78; RRID: AB_2493099 | IF (1:300) |
| Antibody | Anti-PanNav (Mouse monoclonal) | Sigma-Aldrich | Clone: K58/35; RRID: AB_477552 | IF (1:300) |
| Antibody | Anti-Kv1.2 (Mouse monoclonal) | Neuromab | Clone: K14_16; RRID: AB_2296313 | IF (1:500) |
| Antibody | Anti-β1 spectrin (Mouse monoclonal) | Neuromab | Clone: N385/21; RRID: AB_2315815 | IF (1:500) WB (1:1000) |
| Antibody | Anti-actin (Mouse monoclonal) | Millipore | Cat.#: MAB1501; RRID: AB_2223041 | WB (1:4000) |
| Antibody | Anti-β2 spectrin (Mouse monoclonal) | BD Biosciences | Cat.#: 612563; RRID: AB_399854 | IF (1:1000) |
| Antibody | Anti-Tuj1 (Mouse monoclonal) | BioLegend | Cat.#: 801201; RRID: AB_2313773 | IF (1:800) |
| Antibody | Anti-β3 spectrin (Rabbit polyclonal) | Novus | Cat.#: NB110-58346; RRID: AB_877723 | IF (1:1000) |

*Continued on next page*

*Continued*

| Reagent type (species) or resource | Designation | Source or reference | Identifiers | Additional information |
|---|---|---|---|---|
| Antibody | Anti-ATF3 (Rabbit polyclonal) | Santa Cruz | Cat.#: SC-188, RRID: AB_2258513 | IF (1:1000) |
| Antibody | Anti-β4 spectrin NT antibody (Rabbit polyclonal) | PMID: 28123356 | RRID: AB_2315634 | IF (1:50) |
| Antibody | Anti-β4 spectrin SD antibody (chicken polyclonal) | PMID: 28123356 | | IF (1:200) |
| Antibody | Anti-β4 spectrin SD antibody (Rabbit polyclonal) | PMID: 28123356 | | IF (1:500) WB (1:1000) |
| Antibody | Anti-Ankyrin R (Rabbit polyclonal) | PMID: 25362473 | | IF (1:500) |
| Antibody | Anti-Caspr (Rabbit polyclonal) | PMID: 10460258 | RRID: AB_2572297 | IF (1:500) |
| Antibody | Anti-Pan Neurofascin (Chicken polyclonal) | R and D Systems | Cat.#: AF3235; RRID: AB_10890736 | IF (1:500) |
| Sequence-based reagent | Genotyping primer for $Sptbn4^{flox/flox}$ mouse (sense) | PMID: 30226828 | | 5'-GAGCTGCATAAGTTCTTCAGCGATGC-3' |
| Sequence-based reagent | Genotyping primer for $Sptbn4^{flox/flox}$ mouse (anti-sense) | PMID: 30226828 | | 5'-ACCCCATCTCAACTGGCTTTCTTGG-3' |
| Sequence-based reagent | Genotyping primer for $Sptb^{flox/flox}$ mouse (sense) | This paper | | 5'- ACAGAGACAGATGGCCGAAC-3' |
| Sequence-based reagent | Genotyping primer for $Sptb^{flox/flox}$ mouse (anti-sense) | This paper | | 5'-CTCTGGTTCCCAGGAGAGC-3' |
| Sequence-based reagent | Genotyping primer for *Avil-cre* mouse (primer 1) | PMID: 29038243 | | 5'-CCCTGTTCACTGTGAGTAGG-3' |
| Sequence-based reagent | Genotyping primer for *Avil-cre* mouse (primer 2) | PMID: 29038243 | | 5'- AGTATCTGGTAGGTGCTTCCAG-3' |
| Sequence-based reagent | Genotyping primer for *Avil-cre* mouse (primer 3) | PMID: 29038243 | | 5'-GCGATCCCTGAACATGTCCATC-3' |
| Software, algorithm | Fiji | National Institutes of Health | RRID:SCR_002285 | |
| Software, algorithm | Qupath | PMID: 29203879 | | |
| Software, algorithm | Prism | Graph Pad | RRID:SCR_002798 | Version 6 |
| Software, algorithm | pClamp | Molecular Devices | RRID:SCR_011323 | |
| Software, algorithm | MultiClamp | Molecular Devices | | |
| Software, algorithm | Clampfit | Molecular Devices | | |

## Animals

To generate $Sptb^{F/F}$ mice, a targeting construct was made in which loxP sites were designed to flank exon 2 (ENSMUSE00000414599), resulting in disruption of the reading frame upon Cre-mediated excision. A genomic bacterial artificial chromosome containing the *Sptb* locus was obtained and genomic DNA fragments harboring exon 2 of *Sptb* were subcloned in pEasy (a gift from Sankar Ghosh). The targeting vector comprised a long arm and a short arm, with a PGK–neomycin resistance (Neor) selection cassette flanked by two FRT sites and the herpes simplex thymidine kinase gene. The targeting vector was linearized and electroporated into mouse embryonic stem cells (ESCs) (129SV/EV). ES cells were selected for neomycin. Positive clones were identified by PCR. A correctly targeted ESC clone was injected into C57BL/6J blastocysts to produce germ line

transmitting chimeric mice. *Sptbn4*$^{F/F}$ mice were generated as described previously (*Unudurthi et al., 2018*). Both *Sptb*$^{F/F}$ and *Sptbn4*$^{F/F}$ mice were maintained on a mixed C57BL/6 and 129/sv background. Advillin-cre (*Avil-cre*) mice were provided by F. Wang (Duke University, Durham, NC). Both male and female mice were used in our studies. All experiments comply with the National Institutes of Health Guide for the Care and Use of Laboratory Animals and were approved by the Baylor College of Medicine Institutional Animal Care and Use Committee.

## Antibodies

The following monoclonal primary antibodies were purchased from the UC Davis/NIH NeuroMab facility: AnkG (106/36; RRID: AB_10673030), Pan-Nav1 (N419/78; RRID: AB_2493099), Kv1.2 (K14_16; RRID: AB_2296313), β1 spectrin (N385/21; RRID: AB_2315815). Other mouse monoclonal antibodies were sourced as follows: Actin (Millipore MAB1501; RRID: AB_2223041), PanNav (Sigma-Aldrich K58/35; RRID: AB_477552), β2 spectrin (BD Biosciences 612563; RRID: AB_399854), Tuj1 (BioLegend 801201; RRID: AB_2313773). Other rabbit polyclonal antibodies were sourced as follows: β3 spectrin (Novus NB110-58346; RRID: AB_877723), ATF3 (Santa Cruz SC-188, RRID: AB_2258513). The following antibodies were described previously: N terminal-directed rabbit anti-β4 spectrin NT antibody (RRID: AB_2315634) and chicken and rabbit anti-βIV Spectrin SD antibodies (*Yoshimura et al., 2016*); rabbit anti-AnkR (*Ho et al., 2014*); rabbit anti-Caspr (RRID: AB_2572297; (*Rasband et al., 1999*). The chicken anti-Pan Neurofascin was purchased from R and D Systems (AF3235; RRID: AB_10890736). Secondary antibodies were purchased from Thermo Fisher Scientific and Jackson ImmunoResearch Laboratories.

## Tissue lysate preparation and immunoblotting

Whole mouse brains were homogenized in homogenization buffer (0.32M sucrose, 5 mM Na$_3$PO$_4$, 1 mM NaF, 0.5 mM PMSF, 1 mM Na$_3$VO$_4$ and protease inhibitor) on ice. Homogenates were then centrifuged at 700Xg for 10 min at 4°C and supernatant were collected to undergo another centrifugation at 27200Xg for 30 min at 4°C. Pellets were resuspended in homogenization buffer and protein concentration were measured. The samples were resolved by SDS-PAGE, transferred to nitrocellulose membrane, and immunoblotted by antibodies.

## Immunofluorescence microscopy

Mouse tissues were dissected and fixed in 4% PFA for 30 min (nerve) or 45 min (brain) on ice and then immersed in 20% sucrose solution overnight at 4°C. Tissues were then embedded in Tissue-Tek O.C.T compound (Sakura Finetek 4583) and sectioned using a Cryostar NX70 cryostat (Thermo Fisher Scientific). Immunostaining and image captures were performed as previously described (*Huang et al., 2017b*). Measurements of axon diameter, circularity and g-ratio were performed using FIJI (National Institutes of Health) and QuPath (*Bankhead et al., 2017*). Axon circularity was calculated as follows: Circularity = 4*π*(Area of the axon cross section)/(perimeter of the axon cross section)$^2$.

## Transmission electron microscopy (TEM)

Mice were perfused with 2.5% glutaraldehyde and 2.0% PFA in 0.1 M cacodylate buffer, pH 7.4. Dorsal roots were dissected and postfixed overnight in the same fixative as the perfusion. The tissues were then stained in a 0.1% tannic acid solution in 0.1M cacodylate buffer for 20 min. After washing, tissues were postfixed in a 1% osmium tetroxide and 1% ferricyanide solution in 0.1 M cacodylate buffer, pH 7.4, for 1 hr. Tissues were washed and then stained with saturated uranyl acetate for 1 hr. After washing, nerves went through a series of gradual dehydration steps by increasing the percentage of ethanol (50%, 70%, 80%, 90%, 95%, and 100%) and gradually infiltrated with increasing percentage of Spurr's resin (Electron Microscopy Sciences). After infiltration, the tissues were embedded in pure resin. Cross sections were thick sectioned (1 um) and stained with Toluidine blue, then thin sectioned (65 nm) for TEM. The sectioning and electron microscopy were performed in the Baylor College of Medicine Integrated Microscopy Core using a Leica EM UC6 Ultramicrotome and a Hitachi H-7500 TEM, respectively.

## Behavioral tests

The accelerating rotarod test was performed as previously described (*Chang et al., 2010*). Hot plate test, tail-immersion test, and gait analysis were performed as previously described (*Huang et al., 2017b*).

## Compound action potential (CAP) recording

CAP recording of dorsal roots was performed as described previously (*Huang et al., 2017b*). Briefly, dorsal roots were dissected and recorded in a continuously perfused, oxygenated, and temperature-regulated (25°C) chamber containing standard Locke's solution (154 mM NaCl, 5.6 mM KCl, 2 mM $CaCl_2$, 5 mM D-glucose, 10 mM HEPES, pH 7.4). Both ends of the root were drawn into suction electrodes and the responses to a depolarizing current were recorded. CAPs were amplified, digitized, recorded, and analyzed on a laboratory computer using Molecular Devices Axon pClamp, Multi-Clamp, and Clampfit software. Nerve conduction velocity was calculated by dividing the length of the root by the time from stimulation to the peak of the CAP.

## Statistical analysis

Unpaired, two-tailed Student's t-test was performed for statistical analysis unless otherwise indicated. Data were collected and processed randomly and analyzed using GraphPad Prism and Microsoft Excel. No statistical methods or power analysis were used to predetermine sample sizes, but our sample sizes are similar to those reported previously (*Susuki et al., 2013*). Data distribution was assumed to be normal. Experimenters were blinded to genotype in the following experiments: all behavioral experiments comparing $Sptb^{F/F}$ and $Avil\text{-}Cre;Sptb^{F/F}$ mice, all behavioral experiments comparing $Sptbn4^{F/F}$ and $Avil\text{-}Cre; Sptbn4^{F/F}$ mice, all electrophysiology recordings, all analyses of Nav channel fluorescence intensity in $Sptb^{F/F};Sptbn4^{F/F}$ and $Avil\text{-}Cre;Sptb^{F/F}; Sptbn4^{F/F}$ mice, all analyses of ATF3 staining in DRG neurons, and all electron microscopy analyses. Experimenters were not blinded to genotype in the following experiments: behavioral tests using $Sptb^{F/F}; Sptbn4^{F/F}$ and $Avil\text{-}Cre;Sptb^{F/F}; Sptbn4^{F/F}$ mice since the clasping and motor impairments were so obvious, and all immunofluorescence analyses of $Avil\text{-}Cre;Sptb^{F/F}$ and $Avil\text{-}Cre; Sptbn4^{F/F}$ mice since genotypes were obvious from the immunostaining itself. No data points were excluded.

## Acknowledgements

The work reported here was supported by the following research grants: NIH NS044916 (MNR); NIH NS069688 (MNR); and by the Dr. Miriam and Sheldon G Adelson Medical Research Foundation (MNR).

## Additional information

### Funding

| Funder | Grant reference number | Author |
| --- | --- | --- |
| National Institutes of Health | NS044916 | Matthew N Rasband |
| National Institutes of Health | NS069688 | Matthew N Rasband |
| Dr. Miriam and Sheldon G. Adelson Medical Research Foundation | | Matthew N Rasband |

The funders had no role in study design, data collection and interpretation, or the decision to submit the work for publication.

### Author contributions

Cheng-Hsin Liu, Formal analysis, Investigation, Visualization, Writing - original draft, Writing - review and editing; Sharon R Stevens, Investigation, Performed all electrophysiology experiments; Lindsay H Teliska, Investigation, Performed electron microscopy experiments; Michael Stankewich, Resources, Constructed the Sptbn1 floxed mouse line; Peter J Mohler, Thomas J Hund, Resources,

Constructed the Sptbn4 floxed mouse line; Matthew N Rasband, Conceptualization, Resources, Funding acquisition, Methodology, Writing - original draft, Project administration, Writing - review and editing

### Author ORCIDs
Cheng-Hsin Liu http://orcid.org/0000-0002-4582-4551
Matthew N Rasband https://orcid.org/0000-0001-8184-2477

### Ethics
Animal experimentation: This study was performed in strict accordance with the recommendations in the Guide for the Care and Use of Laboratory Animals of the National Institutes of Health. All of the animals were handled according to approved institutional animal care and use committee (IACUC) protocols (AN-4634) at the Baylor College of Medicine.

### Decision letter and Author response
Decision letter https://doi.org/10.7554/eLife.52378.sa1
Author response https://doi.org/10.7554/eLife.52378.sa2

## Additional files

### Supplementary files
• Transparent reporting form

### Data availability
All data generated or analysed during this study are included in the manuscript and supporting files. Source data are included for all results.

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
