## [Decision Letter]

**Acceptance summary:**

In this study, Liu et al. present data suggesting that nodal β spectrins are critical for maintaining nodal Nav clustering and axon integrity. Under conditions of β4 spectrin deletion, peripheral sensory axon function and behavioral function is intact, indicating that the spectrin isoform is not required for Nav clustering at nodes and axon function. However, the deletion of β4 spectrin alone led to an increase in β1 spectrin expression at nodes that had experienced a loss of the β4 spectrin. In addition, because the deletion of the β1 spectrin resulted in no obvious behavioral or axonal abnormalities unless it was paired with β4 spectrin KO, β1 spectrin appears to not play an independent role in maintaining nodal structural integrity and function and may instead participate in a compensatory mechanism by which Nav clustering is maintained in peripheral sensory neurons. Indeed, the authors observed that the loss of the two spectrins led to the progressive reduction of nodal Navs over the course of several months and ushered in ultrastructural axon changes like deformation, abnormal myelination, and changes in axon diameter, in addition to an uptick in the expression of the axon injury marker ATF3. Altogether, the data are compelling and of high quality, and suggest that the two spectrins act with hierarchical contributions to maintain Nav clustering at nodes, in addition to axon integrity.

**Decision letter after peer review:**

Thank you for submitting your article "Nodal β spectrins are required to maintain Na^+^ channel clustering and axon integrity" for consideration by *eLife*. Your article has been reviewed by three peer reviewers, one of whom is a member of our Board of Reviewing Editors, and the evaluation has been overseen by Olga Boudker as the Senior Editor. The following individual involved in review of your submission has agreed to reveal their identity: Vann Bennett (Reviewer #3).

The reviewers have discussed the reviews with one another and the Reviewing Editor has drafted this decision to help you prepare a revised submission. All three reviewers are excited about the findings and found the study suitable for publication in *eLife*. There are a few technical questions that they have raised.

Essential revisions:

While the study provides valuable insight in how nodal architecture contributes to Nav clustering, axon integrity, and resultant motor/sensory abnormalities, there remain a number of issues that should be addressed to strengthen the work.

1) "Maintenance" may not accurately describe Nav channel clustering by the two nodal spectrins, as it is unclear whether the reduction in nodes with Nav channels seen in the double cKO represents a loss of nodes with Nav channels or the decrease in the formation of new nodes. This confound could be readily resolved by the use of an inducible cKO-however, this comment does not really change the conclusions of the manuscript. Therefore, the authors could also just quantify Nav expression at nodes at a couple of additional time points during development to better distinguish a mechanism for maintenance.

2) The authors showed that b1/b4 double knockout in PNS sensory neurons exhibited motor dysfunction and impaired action potential. However, these double mutant mice showed normal Nav clustering until 6 months in age. Since the mice in Figure 4 were only 6 weeks old, there is a discrepancy between the behavior phenotypes and lack of structural deficit. The authors should look at more specific groups of neurons involved in proprioception to see if a subgroup of neurons is selectively affected at early age. This will potentially change their conclusion.

3) In Figure 6D, is it possible that the difference in circularity is due to EM artifact? It is also unclear how the authors calculated axon circularity-this could be discussed.

4) In Figure 6H, it would be useful to see a breakdown of changes in axon diameter relative to control binned by small, medium, and large-diameter ranges. This suggestion is based on previous work from the Rasband lab, which found that spectrin ɑII protects large-diameter neurons from degeneration.

5) In Figure 6, the authors should determine whether or not the degeneration of axons is limited to myelinated axons or whether it also impacts unmyelinated axons, to determine whether or not the loss of "nodal expression" of these two spectrins that leads to axon degeneration. If degeneration were also to occur in unmyelinated axons, then the conclusion needs to be made clear in the manuscript.

---

## [Author Response]

Essential revisions:While the study provides valuable insight in how nodal architecture contributes to Nav clustering, axon integrity, and resultant motor/sensory abnormalities, there remain a number of issues that should be addressed to strengthen the work.1) "Maintenance" may not accurately describe Nav channel clustering by the two nodal spectrins, as it is unclear whether the reduction in nodes with Nav channels seen in the double cKO represents a loss of nodes with Nav channels or the decrease in the formation of new nodes. This confound could be readily resolved by the use of an inducible cKO-however, this comment does not really change the conclusions of the manuscript. Therefore, the authors could also just quantify Nav expression at nodes at a couple of additional time points during development to better distinguish a mechanism for maintenance.

This is a good point and we apologize that we did not provide the total node counts. We reported only the percentage of nodes with Nav channels. Therefore, we add new data in the results reporting no change in the total number of nodes at 1M and 6M of age. A figure summarizing these findings is included as Figure 5—figure supplement 1.

2) The authors showed that b1/b4 double knockout in PNS sensory neurons exhibited motor dysfunction and impaired action potential. However, these double mutant mice showed normal Nav clustering until 6 months in age. Since the mice in Figure 4 were only 6 weeks old, there is a discrepancy between the behavior phenotypes and lack of structural deficit. The authors should look at more specific groups of neurons involved in proprioception to see if a subgroup of neurons is selectively affected at early age. This will potentially change their conclusion.

We respectfully disagree with this conclusion that there is a discrepancy in the results. Although it is correct that the behavioral assays were performed at 6 weeks of age, it is not correct that no structural deficits were observed. As shown in Figure 5B, already at one month of age there was a significant reduction (~30%) in the percentage of nodes with Na^+^ channels. Thus, structural deficits in nodes were detected at 1 month of age, two weeks before we performed the behavioral assays reported in Figure 4. Nevertheless, it is a good point that there may be preferential populations of neurons that are affected. The behavior and physiology results in Figure 4 clearly support a defect in myelinated axons (mostly likely proprioceptor and mechanoreceptors). To begin to define the populations of neurons affected by loss of b1/b4 spectrin we measured the diameter of DRG neurons that are ATF3 positive. We found that large diameter DRG neurons, which correspond to myelinated proprioceptors and mechanoreceptors, were significantly more likely to be ATF3 positive. These results are now included in a revised Figure 6C.

3) In Figure 6D, is it possible that the difference in circularity is due to EM artifact? It is also unclear how the authors calculated axon circularity-this could be discussed.

We can only respond that we performed these experiments in triplicate and all cKO mice showed the same defect. Importantly, in a completely separate study performed 15 years ago on b4 spectrin mutant QV3J mice we also found dramatic changes in the circularity of axons (see Yang et al., 2004). We now include the formula for how axon circularity was calculated in the Materials and methods.

4) In Figure 6H, it would be useful to see a breakdown of changes in axon diameter relative to control binned by small, medium, and large-diameter ranges. This suggestion is based on previous work from the Rasband lab, which found that spectrin ɑII protects large-diameter neurons from degeneration.

This is a good point. We have followed this suggestion and show axon diameters in 1 μm bins (revised Figure 6H). These results show that there is an increase in the number of small diameter axons. Perhaps a more relevant data point to the comment that large diameter axons are more affected by loss of a2 spectrin: as stated above in point #2 (new Figure 6C), ATF3 positive neurons were more likely to be large diameter neurons. This is consistent with our previous report in the a2 spectrin cKO mice (Huang et al., 2017).

5) In Figure 6, the authors should determine whether or not the degeneration of axons is limited to myelinated axons or whether it also impacts unmyelinated axons, to determine whether or not the loss of "nodal expression" of these two spectrins that leads to axon degeneration. If degeneration were also to occur in unmyelinated axons, then the conclusion needs to be made clear in the manuscript.

We do not think that unmyelinated axons are significantly affected by loss of b1/b4 spectrin. First, the tail immersion test which measures function of nociceptive unmyelinated axons was not different from control mice. Second, as reported in point #2, above (Figure 6C), ATF3 positive DRG were large diameter neurons which correspond to myelinated axons.